# 🐱 VIDEO DETECTIVE: SEEK CRITICAL CLUES RECURRENTLY TO ANSWER QUESTION FROM LONG VIDEOS

## ABSTRACT

Long Video Question-Answering (LVQA) presents a significant challenge for Multi-modal Large Language Models (MLLMs) due to immense context and over-loaded information, which could also lead to prohibitive memory consumption. While existing methods attempt to address these issues by reducing visual tokens or extending model's context length, they may miss useful information or take considerable computation. In fact, when answering given questions, only a small amount of crucial information is required. Therefore, we propose an efficient question-aware memory mechanism, enabling MLLMs to recurrently seek these critical clues. Our approach, named VideoDetective, simplifies this task by iteratively processing video sub-segments. For each sub-segment, a question-aware compression strategy is employed by introducing a few special memory tokens to achieve purposefully compression. This allows models to effectively seek critical clues while reducing visual tokens. Then, due to history context could have a significant impact, we recurrently aggregate and store these memory tokens to update history context, which would be reused for subsequent sub-segments. Furthermore, to more effectively measure model's long video understanding ability, we introduce GLVC (Grounding Long Video Clues), a long video question-answering dataset, which features grounding critical and concrete clues scattered throughout entire videos. Experimental results demonstrate our method enables MLLMs with limited context length of $32K$ to efficiently process $100K$ tokens (3600 frames, an hour-long video sampled at $1fps$), requiring only 2 minutes and 37GB GPU memory usage. Evaluation results across multiple long video benchmarks illustrate our method can more effectively seek critical clues from massive information.

## 1 INTRODUCTION

Long video question-answering task aims to answer questions from minutes-long or even hours-long videos, requiring models to process a large number of video frames. Mainstream Multi-modal Large Language Models (MLLMs) represent a single video frame with a substantial number of tokens. For example, Qwen2.5-VL (Bai et al., 2025) represents an image of $224 \times 224$ resolution as $64$ tokens, while LLaVA-NeXT (Zhang et al., 2024b) uses $144$ tokens. Therefore, a ten minutes-long video input (600 frames sampled at $1fps$) could result in at least $38K$ tokens, which easily exceeds the maximum context length of Qwen2.5-VL ($32K$) and causes memory explosion.

Recently, there have been many research works dedicated to solving these problems. They either reduced the number of visual tokens by merging adjacent frames with similar semantics (He et al., 2024; Song et al., 2024; Weng et al., 2024) or extended the context length of MLLMs followed by fine-tuning on the long videos data (Dubey et al., 2024; Bai et al., 2023; Xiong et al., 2023). While these approaches enable handing longer video input, the former may result in the loss of significant information, affecting the understanding of long videos; the latter takes considerable computation and GPU memory costs. Therefore, a more efficient method is urgently needed for long video understanding task.

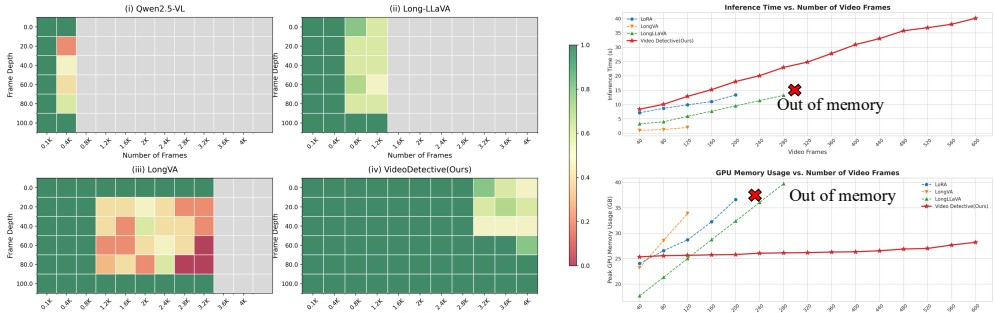

(a) Left: The overview of VideoDetective equipped with an efficient question-aware memory mechanism. Right: Evaluation results on multiple long video benchmarks.

(b) Left: Visual Needle-In-The-Haystack evaluation. The x-axis represents the total number of frames in the video haystack. The y-axis shows the position where the needle image is located. Gray grids mean "OOM". Right: Inference efficiency compared with LoRA and two long video models.

Figure 1: We present VideoDetective, an Multi-modal Large Language Model equipped with efficient question-aware memory mechanism. As shown in Fig. 1(a), it features **recurrently seeking critical clues related to question** from minutes or even hours-long videos. Compared to entire videos, only a few special tokens are used to answer question, thus saving GPU memory usage while effectively leveraging crucial information. Fig. 1(b) shows the Visual Needle-In-The-Haystack evaluation (Zhang et al., 2024a) and inference efficiency. Our proposed efficient question-aware memory mechanism enables models with limited context length, such as Qwen2.5-VL, to efficiently process $4K$ video frames input, requiring only 2 minutes and 37GB GPU memory usage. Moreover, compared to other long video understanding models, VideoDetective could more effectively seek critical "needles" from video haystack, demonstrating superior long video understanding capabilities.

Essentially, a long video contains a massive amount of information, most of which is redundant and irrelevant to the question, Thus, it is a more efficient approach is to seek small amount of crucial clues. For example, as shown in Fig. 1(a), in the cartoon video of "Tom and Jerry", there are three critical clues indicating that the small kitten and Jerry are friends (as marked by the red dashed line). Only combining these clues, the model can correctly answer the question "Why didn't the small kitten catch the mouse?" More intuitively, when we humans complete this task, we usually adopt a strategy of **thinking while watching**, that is, thinking with questions in mind. During this process, we look for important information related to question and remember them in our mind. Then we continue to watch subsequent video content until end. Finally we integrate all collected information in the past to answer question. Compared with watching entire video at once and then thinking about question, this strategy is easier and more efficient.

Inspired by this progressive thinking process, we propose an efficient question-aware memory mechanism, enabling models to recurrently seek critical clues related to question from long videos. Specifically, it consists of two parts: **1) question-aware compression of visual tokens**: we first append a few learnable special tokens (also called memory tokens) to the end of each input video sub-segment. To avoid missing crucial information, the questions are inserted before memory tokens to achieve purposeful compression. Therefore, these memory tokens naturally serve as queries and could aggregate crucial semantic representations due to the causal attention and autoregressive properties of language models (Lester et al., 2021; Kitaev et al., 2020). **2) recurrently seek critical clues**: as history context could also have a significant impact on critical clues, the memory tokens with crucial semantic from each sub-segment are extracted and stored in the memory bank

to update history context. They would be reused as additional inputs when processing subsequent sub-segments. Finally, only extremely few memory tokens are used to answer the question, without relying on any video input, thus saving GPU memory usage while effectively seeking critical clues.

Further, to more effectively measure the model's ability to seek critical clues from long videos, we introduce **GLVC** (**G**rounding **L**ong **V**ideo **C**lues), a long video question-answering dataset that features grounding concrete and critical clues scattered throughout entire long videos. Although there have been many long video benchmarks available, the tasks in them can be completed relying solely on a small set of sampled video frames, and they mainly evaluate model's final prediction results. Different from them, GLVC includes concrete crucial clues and timestamps (as shown in Fig. 4(b)), which can be used to quantitatively assess whether the models truly understand long videos. Experimental results across multiple benchmarks, as shown in Fig. 1(a) (Right) demonstrate our method achieves superior results. Further, the evaluation results on GLVC dataset illustrate it has stronger capability to seek critical clues compared to other long video understanding models.Overall, our contributions can be summarized as follows.

- We propose an efficient question-aware memory mechanism to recurrently seek critical clues related to question from long videos, enabling models with limited context length of $32K$ to efficiently process $100K$ tokens (an hour-long video sampled at $1fps$), only requiring 2 minutes and 37GB GPU memory usage.

- To more effectively measure model's ability to seek critical clues from long context, we introduce GLVC (Grounding Long Video Clues), a long video question-answering dataset. Different from existing long video benchmarks, it features grounding critical and concrete clues scattered throughout entire videos.

- Experimental results across multiple long video benchmarks show that our method can significantly reduce GPU memory usage with comparable inference time while effectively seeking critical clues from long videos, demonstrating great potential on hours-long video understanding tasks.

## 2    RELATED WORK

### 2.1    MLLMS FOR SHORT VIDEO UNDERSTANDING

In the past few years, MLLMs for short video understanding have made significant progress (Cheng et al., 2024; Li et al., 2023; Ataallah et al., 2024; Sun et al., 2024; Lin et al., 2023). VideoLLaMA 2 (Cheng et al., 2024) incorporates a tailor-made spatial-temporal convolution connector to effectively capture the intricate spatial and temporal dynamics of video data. Video-LLaVA (Lin et al., 2023) aligns multi-modal representations before projection and endows LLM with the ability to comprehend both images and videos simultaneously. To obtain fine-grained temporal information required by video understanding, Sun *et al.* (Sun et al., 2024) propose a multi-resolution causal Q-Former structure. These works mainly focus on short video understanding, and the models have difficulty in handling long videos.

### 2.2    LONG VIDEO QUESTION ANSWERING

Originating from long text modeling in natural language processing, some research works (Song et al., 2024; Weng et al., 2024; Li et al., 2024b; Qian et al., 2025) have been attempted to solve the long video question answering task. Song *et al.* (Song et al., 2024) and Weng *et al.* (Weng et al., 2024) merge the most similar tokens in the adjacent frames to reduce token numbers and enable LLMs to process long videos. LLaMA-VID (Li et al., 2024b) represents each frame with two distinct tokens, namely context token and content token to reduce the computational overhead while preserving the critical information. Qian *et al.* (Qian et al., 2025) sequentially encodes video clips and distills condensed representation using a constant number of video tokens. While these works can effectively reduce token numbers, they usually adopt a purposeless compression method, which may lose important information. Unlike them, our method can effectively seek critical information related to question using only few tokens.

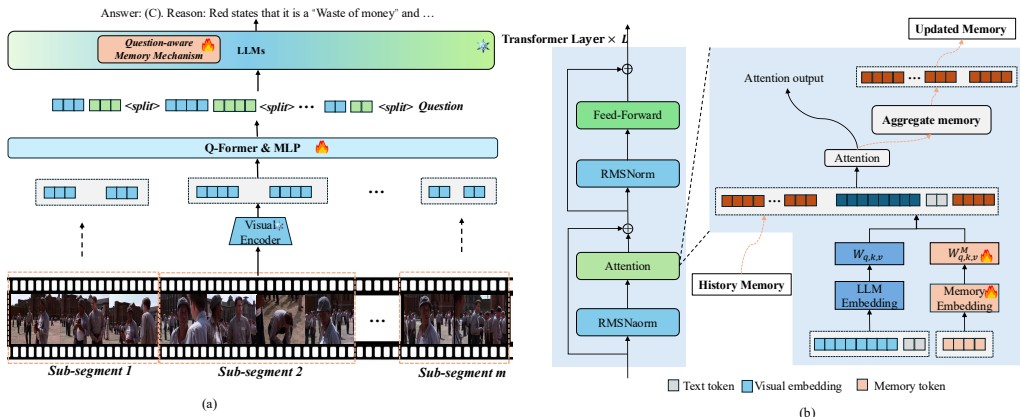

Figure 2: (a) The architecture of our VideoDetective model. The video segment is divided into multiple sub-segments, which are processed by visual encoder to get multi-modal embeddings. Then these embeddings are separated by special `<split>` tokens and the model processes each sub-segment recurrently. (b) The question-aware memory mechanism in attention module at every transformer layer of LLMs. During the process of processing each sub-segment, only few special memory tokens `<memory>` are appended at the end of sub-segment sequence. Then the memory tokens from all past sub-segments and current sub-segment perform attention calculations with other tokens, and then all the memory tokens are aggregated and stored as historical information for subsequent sub-segments.

## 2.3 LONG VIDEO UNDERSTANDING BENCHMARKS

For long video understanding, researchers have proposed numerous benchmarks (Wang et al., 2024a; Zhou et al., 2024; Wu et al., 2024; Fu et al., 2025) from various perspectives to evaluate the model's ability. For example, the LVU dataset (Zhou et al., 2024) collected movie videos with varying lengths and constructed diversified tasks to comprehensively evaluation the model's ability. LongVideoBench (Wu et al., 2024), includes varying-length web-collected videos with subtitles across diverse themes, featuring video-language interleaved inputs up to an hour for longer and richer inputs. However, these benchmarks simply evaluate model's final prediction results, which can not comprehensively prove the model truly understands long videos. In this work, we further propose the GLVC dataset, which could be used to more effectively measure model's ability to seek critical clues from long context.

## 3 METHOD

In this section, we first introduce the model architecture of VideoDetective in Sec. 3.1, then we provide a detailed introduction about question-aware memory mechanism in Sec. 3.2, followed by the training and inference process in Sec. 3.3 .

## 3.1 MODEL ARCHITECTURE

The model architecture of VideoDetective follows a standard design, which includes a visual encoder, a visual-language projector, and an LLM backbone equipped with an efficient question-aware memory mechanism, as shown in Fig. 2(a). Given a question $\mathcal{Q}$ and a long video input $\mathcal{I} = \{I_i \in \mathbb{R}^{H \times W \times C}\}_{i=1}^{T}$, where $H, W, C$ represent the height, width and channels respectively, $T$ is the number of video frames, the model needs to predict the answer, denotes as $\mathcal{A}$. The long video is first divided into $S$ sub-segments $\mathcal{V} = \{V_i\}_{i=1}^{S}$. Then we expand the tokenizer by adding an additional special token `<split>`, which is inserted between adajcent video sub-segments to facilitate distinction. The efficient question-aware memory mechanism is applied in the attention modules of each transformer layer of the LLM backbone, as shown in Fig. 2(b). Next, we will provide a detailed introduction to this module.

## 3.2 EFFICIENT QUESTION-AWARE MEMORY MECHANISM

**Question-aware Compression of Visual Tokens** Due to the causal attention and autoregressive nature, the language models will spontaneously aggregates the sequence information onto the last few tokens (Lester et al., 2021; Kitaev et al., 2020). These tokens naturally serve as a compact representation and provide a high-level summary of current input sequence. Therefore, we can introduce a small set of learnable memory tokens $M = \{<\texttt{memory}>_1, \cdots, <\texttt{memory}>_k\}$, where $k$ is the number of memory tokens. Then these tokens are appended to the end of the video segments to compress visual tokens. To avoid the loss of important information and achieve purposeful compression, we additionally insert the questions into current inputs to facilitate seeking critical clues. Therefore, the inputs are organized as:

$$\underbrace{\{V_1, \mathcal{Q}, M_1\}}_{Seg_1}, <\texttt{split}>, \underbrace{\{V_2, \mathcal{Q}, M_2, \}}_{Seg_2}, <\texttt{split}>, \cdots, \underbrace{\{V_s, \mathcal{Q}, M_s\}}_{Seg_S}, <\texttt{split}>, \mathcal{Q}, \quad (1)$$

where $M_i$ is the memory tokens for video segment $V_i$. Considering the video sub-segments of different lengths may contain varying amounts of information, thus requiring storage of more critical clues, we introduce a compression ratio $\alpha$ to dynamically adjust the number of memory tokens. It can be formalized as $k = N_i/\alpha$, where $N_i$ is the visual token numbers of $V_i$.

**Recurrently Seek Critical Clues** A long video is split into $S$ sub-segments, which are processed recurrently. In each iteration, the model needs to seek and preserve critical semantic representations related to question, which are then input as history context for subsequent processing steps. Specifically, as shown in Fig. 2(b), assume in the $t_{th}$ iteration, in each transformer layer of LLM backbone, the memory tokens $M_t$ for current video segments $V_t$ are first converted into memory embedding, and then multiplied with three trainable matrices $W_{q,k,v}^m$ to obtain $Q^m$, $K^m$, and $V^m$:

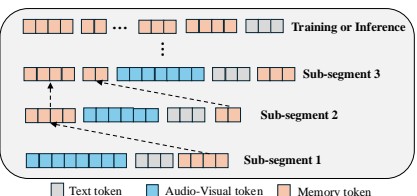

Figure 3: The training and inference process.

$$Q^m, K^m, V^m = W_{q,k,v}^m \cdot E^m(M_t), \quad (2)$$

where $E^m(\cdot)$ is a trainable token embedding similar to word embedding in LLMs. The $Q, K, V$ for video and question embeddings can also be obtained in the same way:

$$Q, K, V = W_{q,k,v} \cdot g([F_v^t \circ F_{\mathcal{Q}}]), \quad (3)$$

where $g(\circ)$ denotes concatenation operation. The $query$, $key$, and $value$ in self-attention calculation can be denoted as:

$$query = g([Q \circ Q^m]), \quad (4)$$

$$key = g([F_k^{past} \circ K \circ K^m]), \quad (5)$$

$$value = g([F_v^{past} \circ V \circ V^m]), \quad (6)$$

where $F_k^{past}, F_v^{past}$ are history context stored in the memory bank from previous iterations. Finally, we aggregate and extract the semantic representations corresponding to memory tokens from results of self-attention computation, which include critical clues related to question. These representations are stored in memory bank to update history context.

Since above process occurs in each transformer layer, the memory bank stores history context from all layers, which are then extracted and reused to each layer in subsequent iterations. In the last step of iteration, the model predicts the answer only based on the history context stored in the memory bank, without relying on any video inputs.

## 3.3 TRAINING AND INFERENCE PROCESS

Fig. 3 demonstrates the overview of training and inference process. As mentioned in sec 3.2, the number of memory tokens will gradually accumulate with the iterations progress. During training process, memory tokens do not participate in loss calculation, and they are only used as an implicit

semantic representation related to question. Therefore, the training loss only consists of the cross-entropy loss:

$$\mathcal{L} = \sum_{j=1}^{l} -log\, p(x_j|V_1, M_1, \mathcal{Q}, \cdots, V_S, M_S, \mathcal{Q}, x_0, ..., x_{j-1}), \tag{7}$$

where $l$ is the token numbers of output $\mathcal{O}$.

During the inference process, all sub-segments are processed in the same way as the training process. After all sub-segments are processed, the question and all memory tokens are used as input to predict the answer, without any video inputs. The next token is predicted as:

$$p(x_j|M_1, ..., M_S, x_0, ..., x_{j-1}), \tag{8}$$

## 4 GLVC DATASET

Evaluating the model's ability to understand long videos is of significant importance. Although there have been many long video benchmarks available, they mainly assess the model's final prediction results, which can not guarantee the model truly understands long videos. Inspired by the Visual Needle-In-The-Haystack evaluation (Zhang et al., 2024a), we further propose the GLVC (**G**rounding **L**ong **V**ideo **C**lues), a long video question-answering dataset that features grounding critical and concrete clues scattered throughout entire videos. In this section, we first introduce the construction pipeline of GLVC dataset in Sec. 4.1 and then provide a detailed analysis in Sec. 4.2.

### 4.1 DATASET CONSTRUCTION PIPELINE

Fig. 4(a) shows the process of dataset construction in detail. Our video data comes from IMDB Top 250 high-rated movies. We first filtered out non-English movies and those before $1960s$. Then we crawled cast images from IMDB website for these movies, and leveraged face detection toolbox[1] and face re-identification model (Chen et al., 2023) to obtain character's images from movie videos. Due to fixed video lengths could disrupt the semantic integrity of plots, we divide the entire movie video into multiple segments based on the scenes. Specifically, we first collected corresponding OCR subtitles from opensubtitles website[2], which includes subtitle data and timestamps. Then we leveraged the powerful reasoning ability of OpenAI o1 model to divide complete segments based on the semantics in the subtitles. These segments are manually checked, corrected, and filtered out inappropriate samples that are too long or too short. Finally, the character images, video segments, and corresponding subtitles are used as inputs. Based on the in-context learning method, we use Gemini 2.5 Pro model (Comanici et al., 2025) to annotate question-answer pairs, concrete reasoning and timestamps for the answer.

### 4.2 DATASET ANALYSIS

The GLVC dataset includes 135 movie videos, with a total duration of 284 hours, covering 20 movie genres. There are a total of $10K$ training samples and $2K$ samples for evaluation. The video lengths vary between $2 \sim 10$ minutes. Clues for each question are scattered throughout the entire video, which can more effectively evaluate the model's long video understanding ability.

The evaluation of model's capabilities includes two parts: 1) The semantic correctness of reasons predicted by the model. 2) The soft temporal IoU between timestamps of critical clues predicted by the model and ground truth. Since the specific timestamps of clues could be a vague and approximate range, we introduce a "tolerance" $\theta$. When the difference between the predicted timestamp of the model and the ground truth is less than $\theta$, the prediction of the model for this timestamp is considered correct. These three metrics can more comprehensively evaluate whether the model truly understands long videos.

---

[1] https://github.com/deepinsight/insightface
[2] https://www.opensubtitles.com/

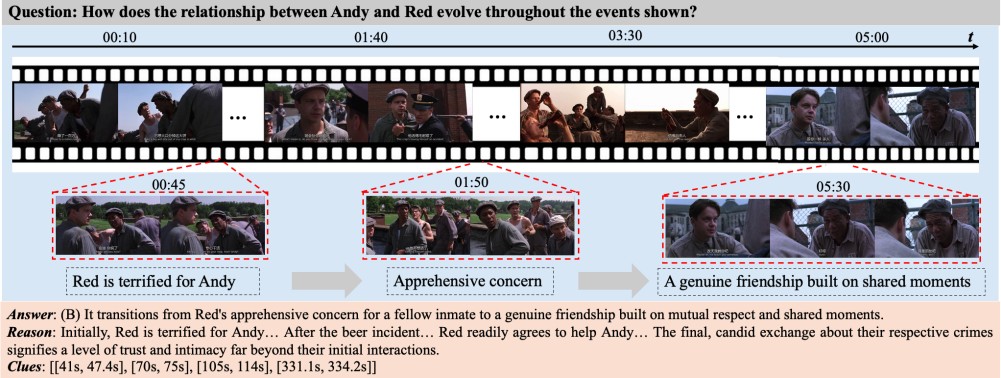

(a) The detailed construction pipeline of GLVC dataset.

(b) A data example from the movie "The Shawshank Redemption" in GLVC dataset.

Figure 4: The overview of GLVC (Grounding Long Video Clues) dataset.

## 5 EXPERIMENTS

### 5.1 IMPLEMENT DETAILS

The video data is sampled at $1fps$ and each frame is resized to $224 \times 224$. The parameters of LLM is initialized from Qwen2.5-VL-7B (Bai et al., 2025). The trainable memory token embedding is initialized with `bos` token embedding of LLMs, and three matrices $W_{q,k,v}^m$ are initialized with the parameters of $W_{q,k,v}$ in LLMs. We employ a two stage training strategy, including warmup and long video training stages.

**Warmup** We found that in the early stage of training, the model has not yet learned to make predictions based on memory tokens for long videos, resulting in high losses and gradients. To maintain training stability, we first warmup the memory-related trainable parameters. Specifically, we collected $8K$ pairs of video-caption data, uniformly sampling up to 60 frames at $1fps$ for each video. Each video segment contains 32 frames, with a fixed compression ratio of 32. The learning rate is $1 \times 10^{-6}$. This warmup stage allows the model to learn to understanding the entire video content using a limited number of memory tokens.

**Long video training** In this stage we collect videos from NExT-QA (Xiao et al., 2021)($32K$), LongVideo-Reason (Chen et al., 2025)($51K$), MovieChat (Song et al., 2024)($1K$), Cinepile (Rawal et al., 2024)($25K$) and GLVC dataset for long video question-answering training with a global batch size 64 and learning rate $1 \times 10^{-4}$. The compression ratio is 16. The video is uniformly sampled up to $512$ frames at $1fps$. Each video segment contains 32 frames.

| Model | Size | VideoMME (w/o sub.) | | | | VideoMME (w/ sub.) | | | | MLVU Test | LVBench | VideoVista | MVBench | Cinepile | NextQA |
|---|---|---|---|---|---|---|---|---|---|---|---|---|---|---|---|
| | | Short | Medium | Long | Avg. | Short | Medium | Long | Avg. | | | | | | |
| **Closed-source Pioneering MLLMs** | | | | | | | | | | | | | | | |
| Gemini-1.5-Pro (Reid et al., 2024) | - | 81.7 | 74.3 | 67.4 | 75.0 | 84.5 | 81.0 | 77.4 | 81.3 | - | 64.0 | - | 60.5 | 60.12 | - |
| GPT-4o (OpenAI, 2024) | - | 80.0 | 70.3 | 65.3 | 71.9 | 82.8 | 76.6 | 72.1 | 77.2 | 54.9 | 66.7 | 78.3 | 64.6 | 56.06 | - |
| Claude 3.5 Sonnet (Anthropic, 2024) | - | 71.0 | 57.4 | 51.2 | 60.0 | 73.5 | 60.1 | 54.7 | 62.9 | - | - | - | - | - | - |
| **Open-source General MLLMs** | | | | | | | | | | | | | | | |
| mPLUG-Owl3 (Ye et al., 2024) | 7B | 70.0 | 57.7 | 50.1 | 59.3 | 72.8 | 66.9 | 64.5 | 68.1 | 17.2 | 59.8 | - | - | 38.27 | - |
| MiniCPM-V 2.6 (Yao et al., 2024) | 8B | 71.3 | 59.4 | 51.8 | 60.9 | 73.5 | 61.1 | 56.3 | 63.7 | - | 54.9 | - | - | 46.91 | - |
| InternVL2 Chen et al. (2024b) | 34B | 72.0 | 59.1 | 52.6 | 61.2 | 72.8 | 61.3 | 53.0 | 62.4 | 45.7 | 59.3 | - | 43.86 | - | - |
| LLaVA-Next-Video (Zhang et al., 2024b) | 34B | 61.7 | 50.1 | 44.3 | 52.0 | 65.1 | 52.2 | 47.2 | 54.9 | 50.5 | - | 56.7 | - | - | - |
| VideoLLaMA 2 (Cheng et al., 2024) | 72B | 69.8 | 59.9 | 57.6 | 62.4 | 72.0 | 63.0 | 59.0 | 64.7 | 45.6 | - | 60.5 | 34.1 | - | - |
| **Open-source Long Video MLLMs** | | | | | | | | | | | | | | | |
| LLaMA-VID Li et al. (2024b) | 7B | - | - | - | - | - | - | - | - | 17.2 | - | 56.9 | 41.4 | - | - |
| LongVA Zhang et al. (2024a) | 7B | 61.1 | 50.4 | 46.2 | 52.6 | 61.6 | 53.6 | 47.6 | 54.3 | 41.1 | - | 67.4 | - | 41.04 | - |
| LongLLaVa Wang et al. (2024b) | 7B | 61.9 | 51.4 | 45.4 | 52.9 | 66.2 | 54.7 | 50.3 | 57.1 | - | - | - | 49.1 | - | - |
| Video-XL Shu et al. (2025) | 7B | 64.0 | 53.2 | 49.2 | 55.5 | 67.4 | 60.7 | 54.9 | 61.0 | 45.5 | 50.7 | 70.6 | 55.3 | - | - |
| LongVILA (Chen et al., 2024a) | 7B | **69.0** | 58.3 | 53.0 | 60.1 | **72.9** | 64.9 | 57.4 | 65.1 | - | 57.1 | - | - | - | 80.7 |
| **VideoDetective (Ours)** | 7B | 68.6 | **61.6** | **56.0** | **62.1** | 70.4 | **66.5** | **63.5** | **66.8** | 45.8 | 51.1 | 74.3 | 58.3 | 67.1* | 79.3 |

Table 1: Experimental results on mainstream short and long video benchmarks. * indicates CinePile training set is included in training data, while results for other models are zero-shot evaluation.

## 5.2 QUANTITATIVE RESULTS AND ANALYSIS

**Evaluation benchmarks** We evaluate VideoDetective's ability on two types of video benchmarks: **1) Long video understanding benchmarks**, including Video-MME (Fu et al., 2024), LongVideoBench (Wu et al., 2024), MLVU (Zhou et al., 2024), VideoVista (Li et al., 2024c) and Cinepile (Rawal et al., 2024). **2) Short video question answering benchmarks**, including MVBench (Li et al., 2024a) and NextQA (Xiao et al., 2021). These benchmarks mainly encompass question-answering tasks in long and short video scenarios.

**Compared models** The compared models mainly include three categories: **1) Closed-source models**: existing pioneering commercial models, which typically have long context lengths and powerful general video understanding capabilities, including Gemini 1.5 Pro (Reid et al., 2024), GPT-4o (OpenAI, 2024) and Claude 3.5 Sonnet (Anthropic, 2024), *etc.*; **2) Open-source general models**, which are usually trained on massive video data and possess general multi-task understanding capability, including mPLUG-Owl3 (Ye et al., 2024), MiniCPM-V 2.6 (Yao et al., 2024), InternVL2 (Chen et al., 2024b), LLaVA-Next-Video (Zhang et al., 2024b) and VideoLLaMA 2 (Cheng et al., 2024), *etc.*. **3) Open-source long video understanding models**, which are specifically designed for understanding long videos, including LLaMA-VID (Li et al., 2024b), LongVA (Zhang et al., 2024a), LongLLaVA (Wang et al., 2024b), Video-XL (Shu et al., 2025) and LongVILA (Chen et al., 2024a), *etc.*.

**Evaluation results across multiple video benchmarks.** We present the performance of VideoDetective on popular long and short video benchmarks in Tab. 1. Compared with closed-source advanced models with massive parameters and powerful general capabilities, there is still some performance gap on these benchmarks. On the long video understanding benchmarks, VideoDetective surpasses most models, including the open-source general models with similar number of parameters (mPLUG-Owl3 and MiniCPM-V 2.6), as well as larger models (34B and 72B). Compared to other long video understanding models with same scale, VideoDetective also demonstrates excellent performance, achieving the best results on the Video MME, MLVU and VideoVista benchmarks, especially for medium and long videos in VideoMME, highlighting VideoDetective's strong capability in understanding long videos relying on seeking critical clues. It is worth noting that on the LongVideoBench, VideoDetective lags behind other models. This is mainly because LongVideoBench features video-language interleved inputs up to an hour long. In our method, each video sub-segment is approximately $30 \sim 60$ seconds long, which would include a significant amount of language content, thus increasing the context length of each sub-segment, making it more challenging to seek clues. VideoDetective also performs well on short video benchmarks. Specifically, it surpasses other open-source models on MVbench and achieves comparable performance on NextQA and the short video testing in VideoMME.

## 5.3 VISUAL NEEDLE-IN-THE-HAYSTACK EVALUATION

To explore VideoDetective's capability in handling long video inputs, we further conducted the Needle-In-The-Haystack evaluation (Zhang et al., 2024a). As shown in Fig. 1(b)(c), we primarily compared the baseline model Qwen2.5-VL and two long video understanding models, LongVA and LongLLava. Since the Qwen2.5-VL model lacks the capability to understand extremely long videos, it encounters memory overflow issues when the number of input frames reaches 800. Although

| Model | Score | | mIoU@5 | | mIoU@10 | | mIoU@15 | |
|---|---|---|---|---|---|---|---|---|
| | w/o sub. | w/ sub. | w/o sub. | w/ sub. | w/o sub. | w/ sub. | w/o sub. | w/ sub. |
| LongLLava Wang et al. (2024b) | 0.47 | 0.56 | 14.5 | 22.5 | 16.7 | 23.3 | 20.8 | 25.6 |
| LongVA Zhang et al. (2024a) | 0.52 | 0.58 | 15.4 | 17.0 | 23.1 | 25.6 | 29.4 | 32.5 |
| **VideoDetective (Ours)** | **0.62** | **0.69** | **23.5** | **26.6** | **36.2** | **37.3** | **42.0** | **44.9** |

Table 2: Evaluation results on GLVC dataset. "Score" represents the semantic similarity between the reasons predicted by model and actual reasons. "mIoU@k" indicates tolerance $\theta = k$ seconds.

LongVA and LongLLava can handle longer video frames, such as $1.2K$ and $3K$, their performance significantly declines. In contrast, VideoDetective, equipped with an efficient question-aware memory mechanism, can process up to $4k$ video frames and still achieve satisfactory results.

**Experimental results on GLVC dataset** Although the Visual Needle-In-The-Haystack evaluation can benchmark model's ability to accurately retrieve information from long context, the "needle" in this evaluation is merely a single image inserted at one position in an hour-long video. To more effectively measure the model's capability to seek critical clues from long videos, we further evaluated the performance of long video models on the GLVC dataset. As shown in Tab. 2, it can be found that compared with LongVA and LongLLaVA models, Video Detective achieves higher GPT scores and mIoU at $\theta$ values of 5s, 10s, and 15s, respectively. This indicates that, with the help of our question-aware memory mechanism, Video Detective can more effectively seek crucial clues. Additionally, subtitle data can also help enhance the model's long video understanding ability.

## 5.4 INFERENCE EFFICIENCY

We further evaluated the inference efficiency of our method compared to LoRA fine-tuning, LongVA and LongLLaVA, including inference time and GPU memory usage. As shown in Fig. 1(b) (Right), our method can process longer videos with comparable inference time but much lower memory usage. Video Detective recurrently processes each seconds-long sub-segment, which does not lead to excessive memory usage while effectively seeking critical information related to question. More importantly, it can process an hour-long video at 1fps (3600 frames) while effectively seeking critical clues, which only requires 2 minutes and 37GB memory usage.

## 5.5 ABLATION RESULTS

To validate effectiveness of our method, we conducted ablation experiments with different compression ratio; As shown in Fig. 5, we evaluated model's performance on MLVU and VideoMME benchmarks with a fixed compression rate 16, 32 and 64, respectively. Experimental results show that when compression rate is fixed, the model's performance gradually decreases as the compression ratio decreases. This is because a larger compression ratio will lead to fewer memory tokens, and thus less information can be sought.

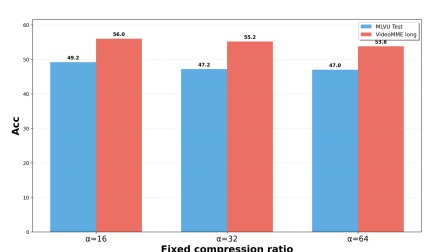

Figure 5: The impact of compression ratio $\alpha$.

## 6 CONCLUSION

In this work, we propose an efficient question-aware memory mechanism, enabling the model to recurrently seek critical clues from long videos. Experimental results on multiple long and short video benchmarks demonstrate the significant potential of our method in understanding long videos. Furthermore, to more effectively evaluate the model's ability to seek critical clues, we have constructed the GLVC dataset, which features concrete clues and allows for the quantitative assessment of the model's long video understanding capabilities.

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

## A  APPENDIX

### A.1  DATASET EXAMPLE

To facilitate readers' understanding of our GLVC dataset, we have provided two additional data examples from the films "The Truman Show" and "Harry Potter and the Deathly Hallows." As shown in Fig. 6, answering the questions in these examples requires collecting sufficient clues from the entire video, rather than relying solely on a few video frames. These clues may be scattered across different corners of the video, allowing for a more comprehensive assessment of a model's ability to understand long videos.

### A.2  ABLATION RESULTS

**Question-aware and history context.** Tab. 3 shows the ablation results on question-aware compression of visual tokens and history context. "*w/o.* question-aware" means the memory tokens do not pay attention to question when processing each sub-segment. The experimental results show that they are an indispensable part of our question-aware memory mechanism. "*w/o.* history context" means previous memory tokens are not reused when processing each sub-segment, and only when predicting answers all the memory tokens are used.

Table 3: Ablation results on history context and question-aware compression of visual tokens.

| Method | MLVU Test | NextQA |
|---|---|---|
| *w/o.* question-aware | 41.5 | 74.2 |
| *w/o.* history context | 43.7 | 75.2 |
| **Video Detective (Ours)** | **45.8** | **79.3** |

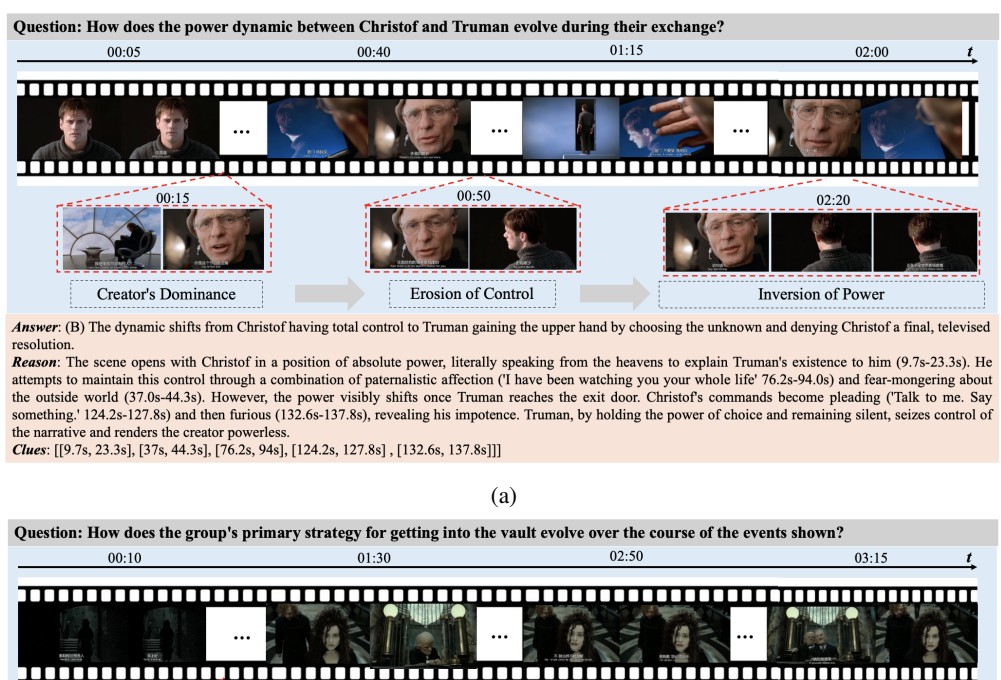

Figure 6: Two data examples in our GLVC dataset comes from the movie "The Truman Show" and "Harry Potter and the Deathly Hallows".

## A.3 LLM USAGE

We employed a large language model (LLM) solely for linguistic refnement of this manuscript,such as grammar correction, phrasing improvement, and style polishing. The LLM was not involvedin research design, data collection, model development, experiments,or analysis, All scientifccontributions, results, and conclusions are entirely the work of the authors.

