# OpenReview forum: "Video Detective: Seek Critical Clues Recurrently to Answer Question from Long Videos"
_ICLR.cc/2026/Conference — Submitted to ICLR 2026_

### Official Review · Reviewer_Y4eZ · 2025-10-25

**Soundness:** 1
**Presentation:** 2
**Contribution:** 1
**Rating:** 2
**Confidence:** 5

**Summary:**

This paper proposes VideoDetective, a multimodal large language model designed to address long video question-answering (LVQA) tasks through an efficient question-aware memory mechanism.

The key contributions include:

  1. Question-aware Memory Mechanism: The method processes long videos by dividing them into sub-segments and employs learnable memory tokens to compress visual information in a
  question-aware manner.
  2. Recurrent Critical Clue Seeking: A memory bank stores and aggregates semantic representations across video segments to maintain historical context for subsequent processing.
  3. GLVC Dataset: A new long video QA dataset featuring concrete temporal clues scattered throughout entire videos, designed to better evaluate models' ability to ground critical
  information.
  4. Efficiency Claims: The method reportedly enables processing of 100K tokens (3600 frames) with only 32K context length, requiring 2 minutes inference time and 37GB GPU memory.

  The approach is motivated by human cognitive processes of "thinking while watching" and aims to seek small amounts of crucial clues rather than processing entire video content at once.

**Strengths:**

**Originality**

  - Dataset Contribution: GLVC provides temporal grounding annotations that could benefit the community for more rigorous evaluation of long video understanding capabilities.

**Quality**

  - Well-Motivated Approach: The method addresses a real limitation of current MLLMs in handling long video contexts due to memory constraints.
  - Comprehensive Evaluation: The paper evaluates on multiple established benchmarks (VideoMME, MLVU, LongVideoBench, etc.) covering both short and long video scenarios.

**Clarity**

  - Clear Problem Statement: The paper clearly articulates the challenges of long video QA and memory limitations.
  - Good Visual Presentation: Figures effectively illustrate the overall architecture and key concepts.

**Significance**

  - Important Problem: Long video understanding is a significant challenge for current MLLMs with practical applications.
  - Memory Efficiency: If the technical claims are validated, the approach could enable broader deployment of video understanding models.

**Weaknesses:**

**Fundamental Training Process Contradictions**
 The paper contains a critical technical inconsistency in Section 3.3:
  - Claims "memory tokens do not participate in loss calculation" yet the loss function $L = \sum_{j=1}^l -\log p(x_j|V_1, M_1, Q, \cdots, V_S, M_S, Q, x_0, ..., x_{j-1})$ explicitly depends
  on memory tokens $M_i$
  - If memory tokens don't participate in loss calculation, how do they receive gradients for optimization?
  - This creates a fundamental contradiction that questions the technical feasibility of the approach

**Gradient Backpropagation Problems**
  The recurrent processing design raises serious gradient flow issues:
  - Each segment's memory tokens depend on historical context from previous segments
  - Maintaining computational graphs across all segments would require enormous memory (contradicting efficiency claims)
  - The paper provides no explanation of how gradients backpropagate through the memory bank updates
  - Missing details on whether detach() operations are used and where

**Incomplete Training Strategy Description**
  The two-stage training process lacks crucial technical details:
  - Warmup stage: Uses video-caption pairs but provides no loss function or training objective
  - Compression ratio inconsistency: Warmup uses ratio 32, main training uses ratio 16 without justification
  - Learning rate jump: 100× increase from warmup (1e-6) to main training (1e-4) lacks theoretical and emperical basis

**Insufficient Ablation Studies**
  Current ablations only test compression ratios, missing critical components:
  - No validation of question-aware compression effectiveness
  - Missing ablation on recurrent memory mechanism vs. simple aggregation
  - No verification that the model actually "seeks critical clues" as claimed

**Unfair Experimental Comparisons**
  - Data leakage: VideoDetective trained on GLVC dataset but evaluated on it (Table 2)

**Sampling Strategy Inconsistencies**
  Section 5.1 reveals problematic data handling:
  - Fixed 32-frame segments ignore semantic boundaries
  - No strategy for handling videos shorter/longer than expected lengths
  - Compression ratio $k = N_i/\alpha$ undefined for segments with $N_i < \alpha$
  - Training-inference mismatch in handling variable-length sequences

**Limited Performance Gains**
  Results show concerning patterns:
  - Large gaps with SOTA: 10-15 point deficits compared to GPT-4o, Gemini-1.5-Pro
  - Failure on key benchmarks: Acknowledged poor performance on LongVideoBench
  - Marginal improvements: Small gains over same-scale models don't justify complexity
  - Short video regression: Performance drops on short videos suggest fundamental limitations

**Questions:**

**Training Process Mechanics**
  - Please provide the complete training algorithm with explicit gradient computation formulas: $\frac{\partial L}{\partial M_i} = ?$ for $i = 1,2,...,S$
  - Explain exactly what "memory tokens do not participate in loss calculation" means technically
  - Provide pseudocode showing how computational graphs are maintained across recurrent segments
  - Clarify the specific loss function and optimization target for the warmup stage

**Experimental Methodology**
  - Can you provide results on GLVC dataset using zero-shot evaluation (without training on GLVC)?
  - What are the exact hardware specifications and software environments for efficiency comparisons?
  - Can you include ablation studies removing question-aware compression and recurrent memory components?

**Sampling and Data Processing**
  - How exactly are videos of different lengths processed during training and inference?
  - What is the strategy for handling the last segment when video length is not divisible by 32?
  - How do you maintain semantic coherence when using fixed-size segments?
  - Can you provide analysis showing the method actually captures "critical clues" rather than random information?

---

### Official Review · Reviewer_mdTa · 2025-10-31

**Soundness:** 3
**Presentation:** 2
**Contribution:** 3
**Rating:** 4
**Confidence:** 4

**Summary:**

This paper aims to reduce token consumption for MLLMs, targeting for long-video understanding. Particularly, VideoDetective method was proposed. VideoDetective segments the long video to short segment. For each segment, learnable memory tokens are obtained as the representation. All the memory tokens for the long video are input to LLM for VQA. It requires much smaller memory consumption than baselines.

**Strengths:**

+ The core idea is driven by a strong insight: "only a small amount of crucial information is required" to answer the question, making the question-aware filtering strategy sound.

+ A dataset GLVC for validating the effectiveness of this method was curated.

+ Good/Competitive performance was achieved.

**Weaknesses:**

1) The memory design is query-dependent. However, when question was changed, the build of memory is need again, reducing the proactiveness of the method.

2) Lack of efficiency metrics: A major claim of the paper is efficiency. However, there are no quantitative results comparing the proposed method's efficiency against baselines. This is critical for an LVQA paper. We need to see metrics like inference time against competitive methods (e.g., sparse attention models or other compression techniques) to validate the "efficient" claim.

3) The training process is not clear. Is the model trained with "next token prediction" objective (7)? If so, are the predicted token the final answer?

4) For the j-th segment, will the history memory tokens be used to learn the memory tokens of this segment?

5) Will the segment length influence the performance a lot?

6) In Table 2, what does sub means in "w/ sub"?

7) The computation cost (how many GPUs are used and how long the training takes) should be discussed.

**Questions:**

See weakness part.

---

### Official Review · Reviewer_rML2 · 2025-11-01

**Soundness:** 3
**Presentation:** 2
**Contribution:** 2
**Rating:** 2
**Confidence:** 5

**Summary:**

This paper focuses the core challenges of long video understanding with VLMs, i.e., the excessive context length, prohibitive memory consumption, and loss of critical information in long context. The authors propose VideoDetective, a framework with recurrent question-aware memory compression and critical clue summarization. For evaluation, the authors introduce GLVC, a dataset with concrete critical clues and timestamps scattered across long videos. The experimental results show improvements on computation efficiency.

**Strengths:**

1. The explored problem is meaningful and the motivation of using recurrent memory compression is clear.
2. The recurrent sub-segment processing limits peak memory usage and friendly to real-time video interaction device deployment.
3. The GLVC benchmark is a contribution to the community for comprehensive evaluation of grounded video understanding.

**Weaknesses:**

1. The authors claim dynamic compression ratio for different sub-segments to avoid over- or under-compression, but the technical details are not presented. The experiments only show results with different fixed compression ratio on different benchmarks.
2. The recurrent memory compression with history memory continuously added to the context is quite similar to [1], with the only difference in question-aware or not. Due to the lack of dynamic compression ratio, the advantage of question-aware compression is not shown in this architecture.
3. The scalability to extremely long videos is doubted. The growing historical memory puts limitation on ultra-long videos in terms of both computation and information forgetting.
4. The presentation is not clear in some crucial technical details. For example, in line 269, the authors claim "memory tokens do not participate in loss calculation", which is quite confusing since the memory tokens are explicitly in the sequence context in loss computation, and how do you optimize the memory representations.

[1] Qian, R., Dong, X., Zhang, P., Zang, Y., Ding, S., Lin, D., & Wang, J. (2024). Streaming long video understanding with large language models. Advances in Neural Information Processing Systems, 37, 119336-119360.

**Questions:**

Why choose to initialize memory tokens with bos token embedding?

---

### Meta-Review · Area_Chair_gvop · 2025-12-24

**Summary:**

Reviewers find the problem important and appreciate the GLVC dataset, but raise major concerns about technical clarity/soundness and evidence: the training description is inconsistent, gradient flow in the recurrent memory setup is unspecified, key training details are missing, ablations do not isolate the claimed benefits, and efficiency claims lack strong quantitative comparisons. Overall leans reject.

**Reviewer Concerns:**

There is no rebuttal provided.

**Reviewer Scores:**

There is no rebuttal provided.

---

### Decision · Program_Chairs · 2026-01-26

Reject